# Semi-Separable Hamiltonian Monte Carlo for Inference in Bayesian Hierarchical Models

**Yichuan Zhang**
School of Informatics
University of Edinburgh
Y.Zhang-60@sms.ed.ac.uk

**Charles Sutton**
School of Informatics
University of Edinburgh
c.sutton@inf.ed.ac.uk

## Abstract

Sampling from hierarchical Bayesian models is often difficult for MCMC methods, because of the strong correlations between the model parameters and the hyperparameters. Recent Riemannian manifold Hamiltonian Monte Carlo (RMHMC) methods have significant potential advantages in this setting, but are computationally expensive. We introduce a new RMHMC method, which we call *semi-separable Hamiltonian Monte Carlo*, which uses a specially designed mass matrix that allows the joint Hamiltonian over model parameters and hyperparameters to decompose into two simpler Hamiltonians. This structure is exploited by a new integrator which we call the *alternating blockwise leapfrog algorithm*. The resulting method can mix faster than simpler Gibbs sampling while being simpler and more efficient than previous instances of RMHMC.

## 1 Introduction

Bayesian statistics provides a natural way to manage model complexity and control overfitting, with modern problems involving complicated models with a large number of parameters. One of the most powerful advantages of the Bayesian approach is hierarchical modeling, which allows partial pooling across a group of datasets, allowing groups with little data to borrow information from similar groups with larger amounts of data. However, such models pose problems for Markov chain Monte Carlo (MCMC) methods, because the joint posterior distribution is often pathological due to strong correlations between the model parameters and the hyperparameters [3]. For example, one of the most powerful MCMC methods is Hamiltonian Monte Carlo (HMC). However, for hierarchical models even the mixing speed of HMC can be unsatisfactory in practice, as has been noted several times in the literature [3, 4, 11]. Riemannian manifold Hamiltonian Monte Carlo (RMHMC) [7] is a recent extension of HMC that aims to efficiently sample from challenging posterior distributions by exploiting local geometric properties of the distribution of interest. However, it is computationally too expensive to be applicable to large scale problems.

In this work, we propose a simplified RMHMC method, called Semi-Separable Hamiltonian Monte Carlo (SSHMC), in which the joint Hamiltonian over parameters and hyperparameters has special structure, which we call *semi-separability*, that allows it to be decomposed into two simpler, separable Hamiltonians. This condition allows for a new efficient algorithm which we call the *alternating blockwise leapfrog algorithm*. Compared to Gibbs sampling, SSHMC can make significantly larger moves in hyperparameter space due to shared terms between the two simple Hamiltonians. Compared to previous RMHMC methods, SSHMC yields simpler and more computationally efficient samplers for many practical Bayesian models.

## 2 Hierarchical Bayesian Models

Let $\mathcal{D} = \{\mathcal{D}_i\}_{i=1}^{N}$ be a collection of data groups where $i$th data group is a collection of iid observations $\mathbf{y}_j = \{y_{ji}\}_{i=1}^{N_i}$ and their inputs $\mathbf{x}_j = \{\mathbf{x}_{ji}\}_{i=1}^{N_i}$. We assume the data follows a parametric

distribution $p(\mathbf{y}_i|\mathbf{x}_i, \boldsymbol{\theta}_i)$, where $\boldsymbol{\theta}_i$ is the model parameter for group $i$. The parameters are assumed to be drawn from a prior $p(\boldsymbol{\theta}_i|\boldsymbol{\phi})$, where $\boldsymbol{\phi}$ is the hyperparameter with a prior distribution $p(\boldsymbol{\phi})$. The joint posterior over model parameters $\boldsymbol{\theta} = (\boldsymbol{\theta}_1, \ldots, \boldsymbol{\theta}_N)$ and hyperparameters $\boldsymbol{\phi}$ is then

$$p(\boldsymbol{\theta}, \boldsymbol{\phi}|\mathcal{D}) \propto \prod_{i=1}^{N} p(\mathbf{y}_i|\mathbf{x}_i, \boldsymbol{\theta}_i)p(\boldsymbol{\theta}_i|\boldsymbol{\phi})p(\boldsymbol{\phi}). \tag{1}$$

This *hierarchical Bayesian* model is popular because the parameters $\boldsymbol{\theta}_i$ for each group are coupled, allowing the groups to share statistical strength. However, this property causes difficulties when approximating the posterior distribution. In the posterior, the model parameters and hyperparameters are strongly correlated. In particular, $\boldsymbol{\phi}$ usually controls the variance of $p(\boldsymbol{\theta}|\boldsymbol{\phi})$ to promote partial pooling, so the variance of $\boldsymbol{\theta}|\boldsymbol{\phi}, \mathcal{D}$ depends strongly on $\boldsymbol{\phi}$. This causes difficulties for many MCMC methods, such as the Gibbs sampler and HMC. An illustrative example of pathological structure in hierarchical models is the Gaussian funnel distribution [11]. Its density function is defined as $p(\mathbf{x}, v) = \prod_{i=1}^{n} \mathcal{N}(x_i|0, e^{-v})\mathcal{N}(v|0, 3^2)$, where $\mathbf{x}$ is the vector of low-level parameters and $v$ is the variance hyperparameter. The pathological correlation between $\mathbf{x}$ and $v$ is illustrated by Figure 1.

## 3 Hamiltonian Monte Carlo on Posterior Manifold

Hamiltonian Monte Carlo (HMC) is a gradient-based MCMC method with auxiliary variables. To generate samples from a target density $\pi(\mathbf{z})$, HMC constructs an ergodic Markov chain with the invariant distribution $\pi(\mathbf{z}, \mathbf{r}) = \pi(\mathbf{z})\pi(\mathbf{r})$, where $\mathbf{r}$ is an auxiliary variable. The most common choice of $\pi(\mathbf{r})$ is a Gaussian distribution $\mathcal{N}(\mathbf{0}, G^{-1})$ with precision matrix $G$. Given the current sample $\mathbf{z}$, the transition kernel of the HMC chain includes three steps: first sample $\mathbf{r} \sim \pi(\mathbf{r})$, second propose a new sample $(\mathbf{z}', \mathbf{r}')$ by simulating the Hamiltonian dynamics and finally accept the proposed sample with probability $\alpha = \min\{1, \pi(\mathbf{z}', \mathbf{r}')/\pi(\mathbf{z}, \mathbf{r})\}$, otherwise leave $\mathbf{z}$ unchanged. The last step is a Metropolis-Hastings (MH) correction. Define $H(\mathbf{z}, \mathbf{r}) := -\log \pi(\mathbf{z}, \mathbf{r})$. The Hamiltonian dynamics is defined by the differential equations $(\dot{\mathbf{z}}, \dot{\mathbf{r}}) = (\partial_{\mathbf{r}}H, -\partial_{\mathbf{z}}H)$, where $\mathbf{z}$ is called the *position* and $\mathbf{r}$ is called the *momentum*.

It is easy to see that $\dot{H}(\mathbf{z}, \mathbf{r}) = \partial_{\mathbf{z}}H\dot{\mathbf{z}} + \partial_{\mathbf{r}}H\dot{\mathbf{r}} = 0$, which is called the energy preservation property [10, 11]. In physics, $H(\mathbf{z}, \mathbf{r})$ is known as the *Hamiltonian energy*, and is decomposed into the sum of the *potential energy* $U(\mathbf{z}) := -\log \pi(\mathbf{z})$ and the *kinetic energy* $K(\mathbf{r}) := -\log \pi(\mathbf{r})$. The most used discretized simulation in HMC is the *leapfrog* algorithm, which is given by the recursion

$$\mathbf{r}(\tau + \epsilon/2) = \mathbf{r}(\tau) - \frac{\epsilon}{2}\nabla_{\mathbf{z}}U(\tau) \tag{2a}$$

$$\mathbf{z}(\tau + \epsilon) = \mathbf{z}(\tau) + \epsilon\nabla_{\mathbf{r}}K(\tau + \epsilon/2) \tag{2b}$$

$$\mathbf{r}(\tau + \epsilon) = \mathbf{r}(\tau + \epsilon/2) - \frac{\epsilon}{2}\nabla_{\boldsymbol{\theta}}U(\tau + \epsilon), \tag{2c}$$

where $\epsilon$ is the step size of discretized simulation time. After $L$ steps from the current sample $(\mathbf{z}(0), \mathbf{r}(0)) = (\mathbf{z}, \mathbf{r})$, the new sample is proposed as the last point $(\mathbf{z}', \mathbf{r}') = (\mathbf{z}(L\epsilon), \mathbf{r}(L\epsilon))$. In Hamiltonian dynamics, the matrix $G$ is called the *mass matrix*. If $G$ is constant w.r.t. $\mathbf{z}$, then $\mathbf{z}$ and $\mathbf{r}$ are independent in $\pi(\mathbf{z}, \mathbf{r})$. In this case we say that $H(\mathbf{z}, \mathbf{r})$ is a *separable* Hamiltonian. In particular, we use the term *standard HMC* to refer to HMC using the identity matrix as $G$. Although HMC methods often outperform other popular MCMC methods, they may mix slowly if there are strong correlations between variables in the target distribution. Neal [11] showed that HMC can mix faster if $G$ is not the identity matrix. Intuitively, such a $G$ acts like a preconditioner. However, if the curvature of $\pi(\mathbf{z})$ varies greatly, a global preconditioner can be inadequate.

For this reason, recent work, notably that on Riemannian manifold HMC (RMHMC) [7], has considered *non-separable* Hamiltonian methods, in which $G(\mathbf{z})$ varies with position $\mathbf{z}$, so that $\mathbf{z}$ and $\mathbf{r}$ are no longer independent in $\pi(\mathbf{z}, \mathbf{r})$. The resulting Hamiltonian $H(\mathbf{z}, \mathbf{r}) = -\log \pi(\mathbf{z}, \mathbf{r})$ is called a *non-separable* Hamiltonian. For example, for Bayesian inference problems, Girolami and Calderhead [7] proposed using the Fisher Information Matrix (FIM) of $\pi(\boldsymbol{\theta})$, which is the metric tensor of posterior manifold. However, for a non-separable Hamiltonian, the simple leapfrog dynamics (2a)-(2c) do not yield a valid MCMC method, as they are no longer reversible. Simulation of general non-separable systems requires the generalized leapfrog integrator (GLI) [7], which requires computing higher order derivatives to solve a system of non-linear differential equations. The computational cost of GLI in general is $\mathcal{O}(d^3)$ where $d$ is the number of parameters, which is prohibitive for large $d$.

In hierarchical models, there are two ways to sample the posterior using HMC. One way is to sample the joint posterior $\pi(\boldsymbol{\theta}, \boldsymbol{\phi})$ directly. The other way is to sample the conditional $\pi(\boldsymbol{\theta}|\boldsymbol{\phi})$ and $\pi(\boldsymbol{\phi}|\boldsymbol{\theta})$, simulating from each conditional distribution using HMC. This strategy is called HMC within Gibbs [11]. In either case, HMC chains tend to mix slowly in hyperparameter space, because the huge variation of potential energy across different hyperparameter values can easily overwhelm the kinetic energy in separable HMC [11]. Hierarchical models also pose a challenge to RMHMC, if we want to sample the model parameters and hyperparameters jointly. In particular, the closed-form FIM of the joint posterior $\pi(\boldsymbol{\theta}, \boldsymbol{\phi})$ is usually unavailable. Due to this problem, even sampling some toy models like the Gaussian funnel using RMHMC becomes challenging. Betancourt [2] proposed a new metric that uses a transformed Hessian matrix of $\pi(\boldsymbol{\theta})$, and Betancourt and Girolami [3] demonstrate the power of this method for efficiently sampling hyperparameters of hierarchical models on some simple benchmarks like Gaussian funnel. However, the transformation requires computing eigendecomposition of the Hessian matrix, which is infeasible in high dimensions.

Because of these technical difficulties, RMHMC for hierarchical models is usually used within a block Gibbs sampling scheme, alternating between $\boldsymbol{\theta}$ and $\boldsymbol{\phi}$. This *RMHMC within Gibbs* strategy is useful because the simulation of the non-separable dynamics for the conditional distributions may have much lower computational cost than that for the joint one. However, as we have discussed, in hierarchical models these variables tend be very strongly correlated, and it is well-known that Gibbs samplers mix slowly in such cases [13]. So, the Gibbs scheme limits the true power of RMHMC.

## 4 Semi-Separable Hamiltonian Monte Carlo

In this section we propose a *non-separable* HMC method that does not have the limitations of Gibbs sampling and that scales to relatively high dimensions, based on a novel property that we will call semi-separability. We introduce new HMC methods that rely on semi-separable Hamiltonians, which we call *semi-separable Hamiltonian Monte Carlo (SSHMC)*.

### 4.1 Semi-Separable Hamiltonian

In this section, we define the semi-separable Hamiltonian system. Our target distribution will be the posterior $\pi(\boldsymbol{\theta}, \boldsymbol{\phi}) = \log p(\boldsymbol{\theta}, \boldsymbol{\phi}|\mathcal{D})$ of a hierarchical model (1), where $\boldsymbol{\theta} \in \mathbb{R}^n$ and $\boldsymbol{\phi} \in \mathbb{R}^m$. Let $\mathbf{r}_{\boldsymbol{\theta}} \in \mathbb{R}^n$ and $\mathbf{r}_{\boldsymbol{\phi}} \in \mathbb{R}^m$ be the momentum variables corresponding to $\boldsymbol{\theta}$ and $\boldsymbol{\phi}$ respectively. The non-separable Hamiltonian is defined as

$$H(\boldsymbol{\theta}, \boldsymbol{\phi}, \mathbf{r}_{\boldsymbol{\theta}}, \mathbf{r}_{\boldsymbol{\phi}}) = U(\boldsymbol{\theta}, \boldsymbol{\phi}) + K(\mathbf{r}_{\boldsymbol{\theta}}, \mathbf{r}_{\boldsymbol{\phi}}|\boldsymbol{\theta}, \boldsymbol{\phi}), \tag{3}$$

where the potential energy is $U(\boldsymbol{\theta}, \boldsymbol{\phi}) = -\log \pi(\boldsymbol{\theta}, \boldsymbol{\phi})$ and the kinetic energy is $K(\mathbf{r}_{\boldsymbol{\theta}}, \mathbf{r}_{\boldsymbol{\phi}}|\boldsymbol{\theta}, \boldsymbol{\phi}) = -\log \mathcal{N}(\mathbf{r}_{\boldsymbol{\theta}}, \mathbf{r}_{\boldsymbol{\phi}}; \mathbf{0}, G(\boldsymbol{\theta}, \boldsymbol{\phi})^{-1})$, which includes the normalization term $\log |G(\boldsymbol{\theta}, \boldsymbol{\phi})|$. The mass matrix $G(\boldsymbol{\theta}, \boldsymbol{\phi})$ can be an arbitrary p.d. matrix. For example, previous work on RMHMC [7] has chosen $G(\boldsymbol{\theta}, \boldsymbol{\phi})$ to be FIM of the joint posterior $\pi(\boldsymbol{\theta}, \boldsymbol{\phi})$, resulting in an HMC method that requires $\mathcal{O}\left((m+n)^3\right)$ time. This limits applications of RMHMC to large scale problems.

To attack these computational challenges, we introduce restrictions on the mass matrix $G(\boldsymbol{\theta}, \boldsymbol{\phi})$ to enable efficient simulation. In particular, we restrict $G(\boldsymbol{\theta}, \boldsymbol{\phi})$ to have the form

$$G(\boldsymbol{\theta}, \boldsymbol{\phi}) = \begin{pmatrix} G_{\boldsymbol{\theta}}(\boldsymbol{\phi}, \mathbf{x}) & \mathbf{0} \\ \mathbf{0} & G_{\boldsymbol{\phi}}(\boldsymbol{\theta}) \end{pmatrix},$$

where $G_{\boldsymbol{\theta}}$ and $G_{\boldsymbol{\phi}}$ are the precision matrices of $\mathbf{r}_{\boldsymbol{\theta}}$ and $\mathbf{r}_{\boldsymbol{\phi}}$, respectively. Importantly, we restrict $G_{\boldsymbol{\theta}}(\boldsymbol{\phi}, \mathbf{x})$ to be independent of $\boldsymbol{\theta}$ and $G_{\boldsymbol{\phi}}(\boldsymbol{\theta})$ to be independent of $\boldsymbol{\phi}$. If $G$ has these properties, we call the resulting Hamiltonian a *semi-separable* Hamiltonian. A semi-separable Hamiltonian is still in general non-separable, as the two random vectors $(\boldsymbol{\theta}, \boldsymbol{\phi})$ and $(\mathbf{r}_{\boldsymbol{\theta}}, \mathbf{r}_{\boldsymbol{\phi}})$ are not independent.

The semi-separability property has important computational advantages. First, because $G$ is block diagonal, the cost of matrix operations reduces from $\mathcal{O}((n+m)^k)$ to $\mathcal{O}(n^k)$. Second, and more important, substituting the restricted mass matrix into (3) results in the potential and kinetic energy:

$$U(\boldsymbol{\theta}, \boldsymbol{\phi}) = -\sum_i \left[\log p(\mathbf{y}_i|\boldsymbol{\theta}_i, \mathbf{x}_i) + \log p(\boldsymbol{\theta}_i|\boldsymbol{\phi})\right] - \log p(\boldsymbol{\phi}), \tag{4}$$

$$K(\mathbf{r}_{\boldsymbol{\theta}}, \mathbf{r}_{\boldsymbol{\phi}}|\boldsymbol{\phi}, \boldsymbol{\theta}) = \frac{1}{2}\left[\mathbf{r}_{\boldsymbol{\theta}}^T G_{\boldsymbol{\theta}}(\mathbf{x}, \boldsymbol{\phi})\mathbf{r}_{\boldsymbol{\theta}} + \mathbf{r}_{\boldsymbol{\phi}}^T G_{\boldsymbol{\phi}}(\boldsymbol{\theta})\mathbf{r}_{\boldsymbol{\phi}} + \log |G_{\boldsymbol{\theta}}(\mathbf{x}, \boldsymbol{\phi})| + \log |G_{\boldsymbol{\phi}}(\boldsymbol{\theta})|\right]. \tag{5}$$

If we fix $(\boldsymbol{\theta}, \mathbf{r}_{\boldsymbol{\theta}})$ or $(\boldsymbol{\phi}, \mathbf{r}_{\boldsymbol{\phi}})$, the non-separable Hamiltonian (3) can be seen as a separable Hamiltonian plus some constant terms. In particular, define the notation

$$A(\mathbf{r}_{\boldsymbol{\theta}}|\boldsymbol{\phi}) = \frac{1}{2}\mathbf{r}_{\boldsymbol{\theta}}^T G_{\boldsymbol{\theta}}(\mathbf{x}, \boldsymbol{\phi})\mathbf{r}_{\boldsymbol{\theta}}, \qquad A(\mathbf{r}_{\boldsymbol{\phi}}|\boldsymbol{\theta}) = \frac{1}{2}\mathbf{r}_{\boldsymbol{\phi}}^T G_{\boldsymbol{\phi}}(\boldsymbol{\theta})\mathbf{r}_{\boldsymbol{\phi}}.$$

Then, considering $(\boldsymbol{\phi}, \mathbf{r}_{\boldsymbol{\phi}})$ as fixed, the non-separable Hamiltonian $H$ in (3) is different from the following separable Hamiltonian

$$
\begin{aligned}
H_1(\boldsymbol{\theta}, \mathbf{r}_{\boldsymbol{\theta}}) &= U_1(\boldsymbol{\theta}|\boldsymbol{\phi}, \mathbf{r}_{\boldsymbol{\phi}}) + K_1(\mathbf{r}_{\boldsymbol{\theta}}|\boldsymbol{\phi}), && (6)\\
U_1(\boldsymbol{\theta}|\boldsymbol{\phi}, \mathbf{r}_{\boldsymbol{\phi}}) &= -\sum_i [\log p(\mathbf{y}_i|\boldsymbol{\theta}_i, \mathbf{x}_i) + \log p(\boldsymbol{\theta}_i|\boldsymbol{\phi})] + A(\mathbf{r}_{\boldsymbol{\phi}}|\boldsymbol{\theta}) + \frac{1}{2}\log|G_{\boldsymbol{\phi}}(\boldsymbol{\theta})|, && (7)\\
K_1(\mathbf{r}_{\boldsymbol{\theta}}|\boldsymbol{\phi}) &= A(\mathbf{r}_{\boldsymbol{\theta}}|\boldsymbol{\phi}) && (8)
\end{aligned}
$$

only by some constant terms that do not depend on $(\boldsymbol{\theta}, \mathbf{r}_{\boldsymbol{\theta}})$. What this means is that any update to $(\boldsymbol{\theta}, \mathbf{r}_{\boldsymbol{\theta}})$ that leaves $H_1$ invariant leaves the joint Hamiltonian $H$ invariant as well. An example is the leapfrog dynamics on $H_1$, where $U_1$ is considered the potential energy, and $K_1$ the kinetic energy.

Similarly, if $(\boldsymbol{\theta}, \mathbf{r}_{\boldsymbol{\theta}})$ are fixed, then $H$ differs from the following separable Hamiltonian

$$
\begin{aligned}
H_2(\boldsymbol{\phi}, \mathbf{r}_{\boldsymbol{\phi}}) &= U_2(\boldsymbol{\phi}|\boldsymbol{\theta}, \mathbf{r}_{\boldsymbol{\theta}}) + K_2(\mathbf{r}_{\boldsymbol{\phi}}|\boldsymbol{\theta}), && (9)\\
U_2(\boldsymbol{\phi}|\boldsymbol{\theta}, \mathbf{r}_{\boldsymbol{\theta}}) &= -\sum_i \log p(\boldsymbol{\theta}_i|\boldsymbol{\phi}) - \log p(\boldsymbol{\phi}) + A(\mathbf{r}_{\boldsymbol{\theta}}|\boldsymbol{\phi}) + \frac{1}{2}\log|G_{\boldsymbol{\theta}}(\mathbf{x}, \boldsymbol{\phi})|, && (10)\\
K_2(\mathbf{r}_{\boldsymbol{\phi}}|\boldsymbol{\theta}) &= A(\mathbf{r}_{\boldsymbol{\phi}}|\boldsymbol{\theta}) && (11)
\end{aligned}
$$

only by terms that are constant with respect to $(\boldsymbol{\phi}, \mathbf{r}_{\boldsymbol{\phi}})$.

Notice that $H_1$ and $H_2$ are coupled by the terms $A(\mathbf{r}_{\boldsymbol{\theta}}|\boldsymbol{\phi})$ and $A(\mathbf{r}_{\boldsymbol{\phi}}|\boldsymbol{\theta})$. Each of these terms appears in the kinetic energy of one of the separable Hamiltonians, but in the potential energy of the other one. We call these terms *auxiliary potentials* because they are potential energy terms introduced by the auxiliary variables. These auxiliary potentials are key to our method (see Section 4.3).

## 4.2 Alternating Block-wise Leapfrog Algorithm

Now we introduce an efficient SSHMC method that exploits the semi-separability property. As described in the previous section, any update to $(\boldsymbol{\theta}, \mathbf{r}_{\boldsymbol{\theta}})$ that leaves $H_1$ invariant also leaves the joint Hamiltonian $H$ invariant, as does any update to $(\boldsymbol{\phi}, \mathbf{r}_{\boldsymbol{\phi}})$ that leaves $H_2$ invariant. So a natural idea is simply to alternate between simulating the Hamiltonian dynamics for $H_1$ and that for $H_2$. Crucially, even though the total Hamiltonian $H$ is not separable in general, both $H_1$ and $H_2$ *are* separable. Therefore when simulating $H_1$ and $H_2$, the simple leapfrog method can be used, and the more complex GLI method is not required.

We call this method the *alternating block-wise leapfrog algorithm* (ABLA), shown in Algorithm 1. In this figure the function "leapfrog"

---
**Algorithm 1** SSHMC by ABLA
---
**Require:** $(\boldsymbol{\theta}, \boldsymbol{\phi})$
  Sample $\mathbf{r}_{\boldsymbol{\theta}} \sim \mathcal{N}(0, G_{\boldsymbol{\theta}}(\boldsymbol{\phi}, \mathbf{x}))$ and $\mathbf{r}_{\boldsymbol{\phi}} \sim \mathcal{N}(0, G_{\boldsymbol{\phi}}(\boldsymbol{\theta}))$
  **for** $l$ in $1, 2, \ldots, L$ **do**
    $(\boldsymbol{\theta}^{(l+\epsilon/2)}, \mathbf{r}_{\boldsymbol{\theta}}^{(l+\epsilon/2)}) \leftarrow \text{leapfrog}(\boldsymbol{\theta}^{(l)}, \mathbf{r}_{\boldsymbol{\theta}}^{(l)}, H_1, \epsilon/2)$
    $(\boldsymbol{\phi}^{(l+\epsilon)}, \mathbf{r}_{\boldsymbol{\phi}}^{(l+\epsilon)}) \leftarrow \text{leapfrog}(\boldsymbol{\phi}^{(l)}, \mathbf{r}_{\boldsymbol{\phi}}^{(l)}, H_2, \epsilon)$
    $(\boldsymbol{\theta}^{(l+\epsilon)}, \mathbf{r}_{\boldsymbol{\theta}}^{(l+\epsilon)}) \leftarrow \text{leapfrog}(\boldsymbol{\theta}^{(l)}, \mathbf{r}_{\boldsymbol{\theta}}^{(l)}, H_1, \epsilon/2)$
  **end for**
  Draw $u \sim \mathcal{U}(0, 1)$
  **if** $u < \min(1, e^{H(\boldsymbol{\theta}, \boldsymbol{\phi}, \mathbf{r}_{\boldsymbol{\theta}}, \mathbf{r}_{\boldsymbol{\phi}}) - H(\boldsymbol{\theta}^{(L\epsilon)}, \boldsymbol{\phi}^{(L\epsilon)}, \mathbf{r}^{(L\epsilon)}, \mathbf{r}_{\boldsymbol{\phi}}^{(L\epsilon)})})$
  **then**
    $(\boldsymbol{\theta}', \boldsymbol{\phi}', \mathbf{r}_{\boldsymbol{\theta}}', \mathbf{r}_{\boldsymbol{\phi}}') \leftarrow (\boldsymbol{\theta}^{(L\epsilon)}, \boldsymbol{\phi}^{(L\epsilon)}, \mathbf{r}_{\boldsymbol{\theta}}^{(L\epsilon)}, \mathbf{r}_{\boldsymbol{\phi}}^{(L\epsilon)})$
  **else**
    $(\boldsymbol{\theta}', \boldsymbol{\phi}', \mathbf{r}_{\boldsymbol{\theta}}', \mathbf{r}_{\boldsymbol{\phi}}') \leftarrow (\boldsymbol{\theta}, \boldsymbol{\phi}, \mathbf{r}_{\boldsymbol{\theta}}, \mathbf{r}_{\boldsymbol{\phi}})$
  **end if**
  **return** $(\boldsymbol{\theta}', \boldsymbol{\phi}')$
---

returns the result of the leapfrog dynamics (2a)-(2c) for the given starting point, Hamiltonian, and step size. We call each iteration of the loop from $1 \ldots L$ an *ABLA step*. For simplicity, we have shown one leapfrog step for $H_1$ and $H_2$ for each ABLA step, but in practice it is useful to use multiple leapfrog steps per ABLA step. ABLA has discretization error due to the leapfrog discretization, so the MH correction is required. If it is possible to simulate $H_1$ and $H_2$ exactly, then $H$ is preserved exactly and there is no need for MH correction.

To show that the SSHMC method by ABLA preserves the distribution $\pi(\boldsymbol{\theta}, \boldsymbol{\phi})$, we also need to show that the ABLA is a time-reversible and volume-preserving transformation in the joint space of $(\boldsymbol{\theta}, \mathbf{r}_{\boldsymbol{\theta}}, \boldsymbol{\phi}, \mathbf{r}_{\boldsymbol{\phi}})$. Let $\mathcal{X} = \mathcal{X}_{\boldsymbol{\theta}, \mathbf{r}_{\boldsymbol{\theta}}} \times \mathcal{X}_{\boldsymbol{\phi}, \mathbf{r}_{\boldsymbol{\phi}}}$ where $(\boldsymbol{\theta}, \mathbf{r}_{\boldsymbol{\theta}}) \in \mathcal{X}_{\boldsymbol{\theta}, \mathbf{r}_{\boldsymbol{\theta}}}$ and $(\boldsymbol{\phi}, \mathbf{r}_{\boldsymbol{\phi}}) \in \mathcal{X}_{\boldsymbol{\phi}, \mathbf{r}_{\boldsymbol{\phi}}}$. Obviously, any reversible and volume-preserving transformation in a subspace of $\mathcal{X}$ is also reversible and volume-preserving in $\mathcal{X}$. It is easy to see that each leapfrog step in the ABLA algorithm is reversible and volume-preserving in either $\mathcal{X}_{\boldsymbol{\theta}, \mathbf{r}_{\boldsymbol{\theta}}}$ or $\mathcal{X}_{\boldsymbol{\phi}, \mathbf{r}_{\boldsymbol{\phi}}}$. One more property of integrator of interest is

*symplecticity*. Because each leapfrog integrator is symplectic in a subspace of $\mathcal{X}$ [10], they are also symplectic in $\mathcal{X}$. Then because ABLA is a composition of symplectic leapfrog integrators, and the composition of symplectic transformations is symplectic, we know ABLA is symplectic.

We emphasize that ABLA is actually *not* a discretized simulation of the semi-separable Hamiltonian system $H$, that is, if starting at a point $(\boldsymbol{\theta}, \mathbf{r}_{\boldsymbol{\theta}}, \boldsymbol{\phi}, \mathbf{r}_{\boldsymbol{\phi}})$ in the joint space, we run the exact Hamiltonian dynamics for $H$ for a length of time $L$, the resulting point will not be the same as that returned by ABLA at time $L$ even if the discretized time step is infinitely small. For example, ABLA simulates $H_1$ with step size $\epsilon_1$ and $H_2$ with step size $\epsilon_2$ where $\epsilon_1 = 2\epsilon_2$, when $\epsilon_2 \to 0$ that preserves $H$.

### 4.3 Connection to Other Methods

Although the SSHMC method may seem similar to RMHMC within Gibbs (RMHMCWG), SSHMC is actually very different. The difference is in the last two terms of (7) and (10); if these are omitted from SSHMC and the Hamiltonians for $\pi(\boldsymbol{\theta}|\boldsymbol{\phi})$, then we obtain HMC within Gibbs. Particularly important among these two terms is the auxiliary potential, because it allows each of the separable Hamiltonian systems to *borrow energy* from the other one. For example, if the previous leapfrog step increases the kinetic energy $K_1(\mathbf{r}_{\boldsymbol{\theta}}|\boldsymbol{\phi})$ in $H_1(\boldsymbol{\theta}, \mathbf{r}_{\boldsymbol{\theta}})$, then, in the next leapfrog step for $H_2(\boldsymbol{\phi}, \mathbf{r}_{\boldsymbol{\phi}})$, we see that $\boldsymbol{\phi}$ will have greater *potential* energy $U_2(\boldsymbol{\phi}|\boldsymbol{\theta}, \mathbf{r}_{\boldsymbol{\theta}})$, because the auxiliary potential $A(\mathbf{r}_{\boldsymbol{\theta}}|\boldsymbol{\phi})$ is shared. That allows the leapfrog step to accommodate a larger change of $\log p(\boldsymbol{\phi}|\boldsymbol{\theta})$ using $A(\mathbf{r}_{\boldsymbol{\theta}}|\boldsymbol{\phi})$. So, the chain will mix faster in $\mathcal{X}_{\boldsymbol{\phi}}$. By the symmetry of $\boldsymbol{\theta}$ and $\boldsymbol{\phi}$, the auxiliary potential will also accelerate the mixing in $\mathcal{X}_{\boldsymbol{\theta}}$.

Another way to see this is that the dynamics in RMHMCWG for $(\mathbf{r}_{\boldsymbol{\phi}}, \boldsymbol{\phi})$ preserves the distribution $\pi(\boldsymbol{\theta}, \mathbf{r}_{\boldsymbol{\phi}}, \boldsymbol{\phi}) = \pi(\boldsymbol{\theta}, \boldsymbol{\phi})\mathcal{N}(\mathbf{r}_{\boldsymbol{\phi}}; \mathbf{0}, G_{\boldsymbol{\phi}}(\boldsymbol{\phi})^{-1})$ but not the joint $\pi(\boldsymbol{\theta}, \boldsymbol{\phi}, \mathbf{r}_{\boldsymbol{\theta}}, \mathbf{r}_{\boldsymbol{\phi}})$. That is because the Gibbs sampler does not take into account the effect of $\boldsymbol{\phi}$ on $\mathbf{r}_{\boldsymbol{\theta}}$. In other words, the Gibbs step has the stationary distribution $\pi(\boldsymbol{\phi}, \mathbf{r}_{\boldsymbol{\phi}}|\boldsymbol{\theta})$ rather than $\pi(\boldsymbol{\phi}, \mathbf{r}_{\boldsymbol{\phi}}|\boldsymbol{\theta}, \mathbf{r}_{\boldsymbol{\theta}})$. The difference between the two is the auxiliary potential. In contrast, the SSHMC methods preserve the Hamiltonian of $\pi(\boldsymbol{\theta}, \boldsymbol{\phi}, \mathbf{r}_{\boldsymbol{\theta}}, \mathbf{r}_{\boldsymbol{\phi}})$.

### 4.4 Choice of Mass Matrix

The choice of $G_{\boldsymbol{\theta}}$ and $G_{\boldsymbol{\phi}}$ in SSHMC is usually similar to RMHMCWG. If the Hessian matrix of $-\log p(\boldsymbol{\theta}|\mathbf{y}, \mathbf{x}, \boldsymbol{\phi})$ is independent of $\boldsymbol{\theta}$ and always p.d., it is natural to define $G_{\boldsymbol{\theta}}$ as the inverse of the Hessian matrix. However, for some popular models, e.g., logistic regression, the Hessian matrix of the likelihood function depends on the parameters $\boldsymbol{\theta}$. In this case, one can use any approximate Hessian $B$, like the Hessian at the mode, and define $G_{\boldsymbol{\theta}} := (B + B(\boldsymbol{\phi}))^{-1}$, where $B(\boldsymbol{\phi})$ is the Hessian of the prior distribution. Such a rough approximation is usually good enough to improve the mixing speed, because the main difficulty is the correlation between model parameters and hyperparameters.

In general, because the computational bottleneck in HMC and SSHMC is computing the gradient of the target distribution, both methods have the same computational complexity $\mathcal{O}(lg)$, where $g$ is the cost of computing the gradient and $l$ is the total number of leapfrog steps per iteration. However, in practice we find it very beneficial to use multiple steps in each blockwise leapfrog update in ABLA; this can cause SSHMC to require more time than HMC. Also, depending on the mass matrix $G_{\boldsymbol{\theta}}$, the cost of leapfrog a step in ABLA may be different from those in standard HMC. For some choices of $G_{\boldsymbol{\theta}}$, the leapfrog step in ABLA can be even faster than one leapfrog step of HMC. For example, in many models the computational bottleneck is the gradient $\nabla_{\boldsymbol{\phi}} \log Z(\boldsymbol{\phi})$, $Z(\boldsymbol{\phi})$ is the normalization in prior. Recall that $G_{\boldsymbol{\theta}}$ is a function of $\boldsymbol{\phi}$. If $|G_{\boldsymbol{\theta}}| = Z(\boldsymbol{\phi})^{-1}$, $Z(\boldsymbol{\phi})$ will be canceled out, avoiding computation of $\nabla_{\boldsymbol{\phi}} \log Z(\boldsymbol{\phi})$. One example is using $G_{\mathbf{x}} = e^v \mathbf{I}$ in Gaussian funnel distribution aforementioned in Section 2. A potential problem of such $G_{\boldsymbol{\theta}}$ is that the curvature of the likelihood function $p(\mathcal{D}|\boldsymbol{\theta})$ is ignored. But when the data in each group is sparse and the parameters $\boldsymbol{\theta}$ are strongly correlated, this $G_{\boldsymbol{\theta}}$ can give nearly optimal mixing speed and make SSHMC much faster.

In general, any choice of $G_{\boldsymbol{\theta}}$ and $G_{\boldsymbol{\phi}}$ that would be valid for separable HMC with Gibbs is also valid for SSHMC.

## 5 Experimental Results

In this section, we compare the performance of SSHMC with the standard HMC and RMHMC within Gibbs [7] on four benchmark models.[1] The step size of all methods are manually tuned so

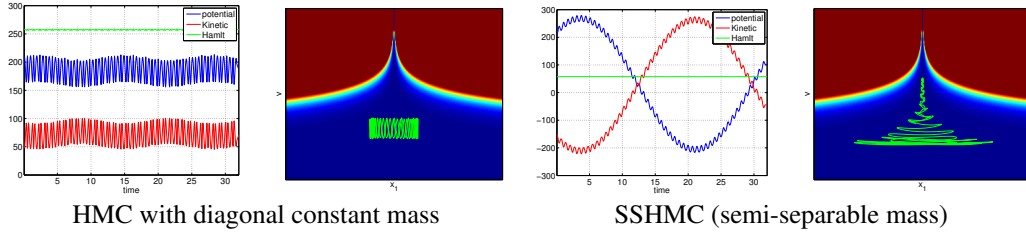

| HMC with diagonal constant mass | SSHMC (semi-separable mass) |
|---|---|

Figure 1: The trace of energy over the simulation time and the trajectory of the first dimension of 100 dimensional Gaussian $\mathbf{x}_1$ (vertical axis) and hyperparameter $v$ (horizontal axis). The two simulations start with the same initial point sampled from the Gaussian Funnel.

| | time(s) | min ESS($\mathbf{x}$, $v$) | min ESS/s ($\mathbf{x}$, $v$) | MSE($\mathbb{E}[v]$, $\mathbb{E}[v^2]$) |
|---|---|---|---|---|
| HMC | 36.63 | (115.35, 38.96) | (3.14, 1.06) | (0.6, 0.18) |
| RMHMC(Gibbs) | 18.92 | (1054.33, 31.69) | (**55.15**, 1.6) | (1.58, 0.72) |
| SSHMC | 22.12 | (**3868.79**, **1541.67**) | (103.57, **41.27**) | (**0.04**, **0.03**) |

Table 1: The result of ESS of 5000 samples on 100 + 1 dimensional Gaussian Funnel distribution. $\mathbf{x}$ are model parameters and $v$ is the hyperparameter. The last column is the mean squared error of the sample estimated mean and variance of the hyperparameter.

| | running time(s) | ESS $\boldsymbol{\theta}$ (min, med, max) | ESS $v$ | min ESS/s |
|---|---|---|---|---|
| HMC | 378 | (2.05, 3.68, 4.79) $\times 10^3$ | 815 | 2.15 |
| RMHMC(Gibbs) | 411 | (0.8, **4.08**, **4.99**)$\times 10^3$ | 271 | 0.6 |
| SSHMC | 385.82 | (**2.5**, 3.42, 4.27)$\times 10^3$ | **2266** | **5.83** |

Table 2: The results of ESS of 5000 samples after 1000 burn-in on Hierarchical Bayesian Logistic Regression. $\boldsymbol{\theta}$ are 200 dimensional model parameters and $v$ is the hyperparameter.

| | time (s) | ESS $\mathbf{x}$(min, med, max) | ESS($\beta, \sigma, \phi$) | min ESS/s |
|---|---|---|---|---|
| HMC | 162 | (1.6, 2.2, 5.2)$\times 10^2$ | (50, 50, 128) | 0.31 |
| RMHMC(Gibbs) | 183 | (12.1, 18.4, 33.5)$\times 10^2$ | (385, 163, 411) | 0.89 |
| SSHMC | 883 | (**78.4**, **98.9**, **120.7**)$\times 10^2$ | (**4434**, **1706**, **1390**) | **1.57** |

Table 3: The ESS of 20000 posterior samples of Stochastic Volatility after 10000 burn-in. $\mathbf{x}$ are latent volatilities over 2000 time lags and $(\beta, \sigma, \phi)$ are hyperparameters. Min ESS/s is the lowest ESS over all parameters normalized by running time.

that the acceptance rate is around 70-85%. The number of leapfrog steps are tuned for each method using preliminary runs. The implementation of RMHMC we used is from [7]. The running time is wall-clock time measured after burn-in. The performance is evaluated by the minimum Effective Sample Size (ESS) over all dimensions (see [6]). When considering the different computational complexity of methods, our main efficiency metric is time normalized ESS.

## 5.1 Demonstration on Gaussian Funnel

We demonstrate SSHMC by sampling the Gaussian Funnel (GF) defined in Section 2. We consider $n = 100$ dimensional low-level parameters $\mathbf{x}$ and 1 hyperparameter $v$. RMHMC within Gibbs on GF has block diagonal mass matrix defined as $G_{\mathbf{x}} = -\partial_v^2 \log p(x, v)^{-1} = e^v \mathbf{I}$ and $G_v = -\mathbb{E}_{\mathbf{x}}[\partial_v^2 \log p(x, v)]^{-1} = (n + \frac{1}{9})^{-1}$. We use the same mass matrix in SSHMC, because it is semi-separable. We use 2 leapfrog steps for low-level parameters and 1 leapfrog step for the hyperparameter in ABLA and the same leapfrog step size for the two separable Hamiltonians.

We generate 5000 samples from each method after 1000 burn-in iterations. The ESS per second (ESS/s) and mean squared error (MSE) of the sample estimated mean and variance of the hyperparameter are given in Table 1. Notice that RMHMC within Gibbs is much more efficient for the low-level variables because the mass matrix adapts with the hyperparameter. Figure 1 illustrates a dramatic difference between HMC and SSHMC. It is clear that HMC suffers from oscillation of the hyperparameter in a narrow region. That is because the kinetic energy limits the change of hyperparameters [3, 11]. In contrast, SSHMC has much wider energy variation and the trajectory spans

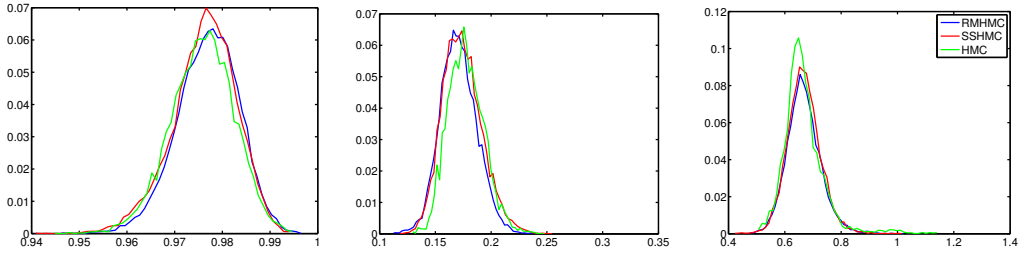

Figure 2: The normalized histogram of 20000 posterior samples of hyperparameters of the stochastic volatility model (from left to right $\phi, \sigma, \beta$) after 10000 burn-in samples. The data is generated by the hyperparameter ($\phi = 0.98, \sigma = 0.15, \beta = 0.65$). All three methods produce accurate estimates, but SSHMC and RMHMC within Gibbs converge faster than HMC.

a larger range of hyperparameter $v$. The energy variation of SSHMC is similar to the RMHMC with Soft-Abs metric (RMHMC-Soft-Abs) reported in [2], an instance of general RMHMC without Gibbs. But compared with [2], each ABLA step is about 100 times faster than each generalized leapfrog step and SSHMC can generate around *2.5 times* more effective samples per second than RMHMC-Soft-Abs. Although RMHMC within Gibbs has better ESS/s on the low level variables, its estimation of the mean and variance is biased, indicating that the chain has not yet mixed. More important, Table 1 shows that the samples generated by SSHMC give nearly unbiased estimates of the mean and variance of the hyperparameter, which neither of the other methods are able to do.

## 5.2  Hierarchical Bayesian Logistic Regression

In this experiment, we consider hierarchical Bayesian logistic regression with an exponential prior for the variance hyperparameter $v$, that is

$$p(\mathbf{w}, \phi | \mathcal{D}) \propto \prod_i \prod_j \sigma(y_{ij} \mathbf{w}_i^T \mathbf{x}_{ij}) \mathcal{N}(\mathbf{w}_i | \mathbf{0}, v\mathbf{I}) \text{Exp}(v | \lambda),$$

where $\sigma$ is the logistic function $\sigma(z) = 1/(1+\exp(-z))$ and $(y_{ij}, \mathbf{x}_{ij})$ is the $j$th data point in the $i$th group. We use the Statlog (German credit) dataset from [1]. This dataset includes 1000 data points and each data has 16 categorical features and 4 numeric features. Bayesian logistic regression on this dataset has been considered as a benchmark for HMC [7, 8], but the previous work uses only one group in their experiments. To make the problem more interesting, we partition the dataset into 10 groups according to the feature *Purpose*. The size of group varies from 9 to 285. There are 200 model parameters (20 parameters for each group) and 1 hyperparameter.

We consider the reparameterization of the hyperparameter $\gamma = \log v$. For RMHMC within Gibbs, the mass matrix for group $i$ is $G_i := \mathcal{I}(\mathbf{x}, \boldsymbol{\theta})^{-1}$, where $\mathcal{I}(\mathbf{x}, \boldsymbol{\theta})$ is the Fisher Information matrix for model parameter $\mathbf{w}_i$ and constant mass $G_v$. In each iteration of the Gibbs sampler, each $\mathbf{w}_i$ is sampled from by RMHMC using 6 generalized leapfrog steps and $v$ is sampled using 6 leapfrog steps. For SSHMC, $G_i := \text{Cov}(\mathbf{x}) + \exp(\gamma)\mathbf{I}$ and the same constant mass $G_v$.

The results are shown in Table 2. SSHMC again has much higher ESS/s than the other methods.

## 5.3  Stochastic Volatility

A stochastic volatility model we consider is studied in [9], in which the latent volatilities are modeled by an auto-regressive AR(1) process such that the observations are $y_t = \epsilon_t \beta \exp(x_t/2)$ with latent variable $x_{t+1} = \phi x_t + \eta_{t+1}$. We consider the distributions $x_1 \sim \mathcal{N}(0, \sigma^2/(1 - \phi^2))$, $\epsilon_t \sim \mathcal{N}(0, 1)$ and $\eta_t \sim (0, \sigma^2)$. The joint probability is defined as

$$p(\mathbf{y}, \mathbf{x}, \beta, \phi, \sigma) = \prod_{t=1}^T p(y_t | x_t, \beta) p(x_1) \prod_{t=2}^T p(x_t | x_{t-1}, \phi, \sigma) \pi(\beta) \pi(\phi) \phi(\sigma),$$

where the prior $\pi(\beta) \propto 1/\beta$, $\sigma^2 \sim \text{Inv-}\chi^2(10, 0.05)$ and $(\phi + 1)/2 \sim \text{beta}(20, 1.5)$. The FIM of $p(\mathbf{x} | \alpha, \beta, \phi, \mathbf{y})$ depends on the hyperparameters but not $\mathbf{x}$, but the FIM of $p(\alpha, \beta, \phi | \mathbf{x}, \mathbf{y})$ depends on $(\alpha, \beta, \phi)$. For RMHMC within Gibbs we consider FIM as the metric tensor following [7]. For SSHMC, we define $G_{\boldsymbol{\theta}}$ as the inverse Hessian of $\log p(\mathbf{x} | \alpha, \beta, \phi, \mathbf{y})$, but $G_{\boldsymbol{\phi}}$ as an identity matrix. In each ABLA step, we use 5 leapfrog steps for updates of $\mathbf{x}$ and 2 leapfrog steps for updates of the hyperparameters, so that the running time of SSHMC is about 7 times that of standard HMC.

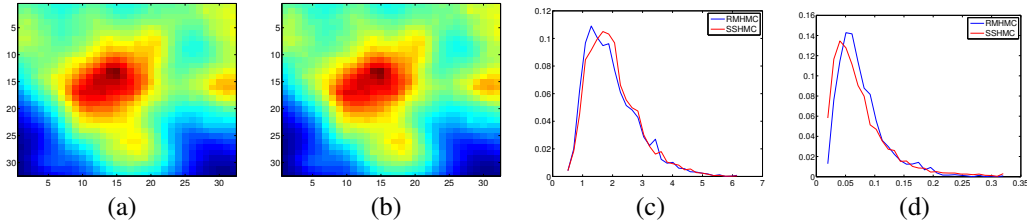

$$\begin{matrix} \text{(a)} & \text{(b)} & \text{(c)} & \text{(d)} \end{matrix}$$

Figure 3: Sample mean of latent fields of the LGCPP model from (a) RMHMC and (b) SSHMC. The normalized histogram of sampled hyperparameter (c) $\sigma$ and (d) $\beta$. We draw 5000 samples from both methods after 1000 burn-in. The true hyperparameter values are ($\sigma = 1.9, \beta = 0.03$).

|  | time(h) | ESS **x**(min, med, max) | ESS($\sigma, \beta$) | min ESS/h |
|---|---|---|---|---|
| SSHMC | 2.6 | (**7.8**, **30**, **39**)$\times 10^2$ | (**2101**, **270**) | **103.8** |
| RMHMC(Gibbs) | 2.64 | (1, 29, 38.3)$\times 10^2$ | (200, 46) | 16 |

Table 4: The ESS of 5000 posterior samples from 32x32 LGCPP after 1000 burn-in samples. **x** is the 1024 dimensional vector of latent variables and ($\sigma, \beta$) are the hyperparameters of the Gaussian Process prior. "min ESS/h" means minimum ESS per hour.

We generate 20000 samples using each method after 10000 burn-in samples. As shown in Figure 2, the histogram of hyperparameters by all methods converge to the same distribution, so all methods are mixing well. But from Table 3, we see that SSHMC generates almost *two times* as many ESS/s as RMHMC within Gibbs.

### 5.4 Log-Gaussian Cox Point Process

The log-Gaussian Cox Point Process (LGCPP) is another popular testing benchmark [5, 7, 14]. We follow the experimental setting of Girolami and Calderhead [7]. The observations $\mathbf{Y} = \{y_{ij}\}$ are counts at the location $(i, j)$, $i, j = 1, \dots, d$ on a regular spatial grid, which are conditionally independent given a latent intensity process $\mathbf{\Lambda} = \{\lambda(i, j)\}$ with mean $m\lambda(i, j) = m \exp(x_{i,j})$, where $m = 1/d^2$, $\mathbf{X} = \{x_{i,j}\}$, $\mathbf{x} = \text{Vec}(\mathbf{X})$ and $\mathbf{y} = \text{Vec}(\mathbf{Y})$. $\mathbf{X}$ is assigned a Gaussian process prior, with mean function $m(x_{i,j}) = \mu\mathbf{1}$ and covariance function $\Sigma(x_{i,j}, x_{i',j'}) = \sigma^2 \exp(-\delta(i, i', j, j')/\beta d)$ where $\delta(\cdot)$ is the Euclidean distance between $(i, j)$ and $(i', j')$. The log joint probability is given by $\log p(\mathbf{y}, \mathbf{x}|\mu, \sigma, \beta) = \sum_{i,j} y_{i,j} x_{i,j} - m \exp(x_{i,j}) - \frac{1}{2}(\mathbf{x} - \mu\mathbf{1})^T \Sigma^{-1}(\mathbf{x} - \mu\mathbf{1})$. We consider a $32 \times 32$ grid that has 1024 latent variables. Each latent variable $x_{i,j}$ corresponds to a single observation $y_{i,j}$.

We consider RMHMC within Gibbs with FIM of the conditional posteriors. See [7] for the FIM for this model. The generalized leapfrog steps are required for updating $(\sigma, \beta)$, but only the leapfrog steps are required for updating $\mathbf{x}$. Each Gibbs iteration takes 20 leapfrog steps for $\mathbf{x}$ and 1 general leapfrog step for $(\sigma, \beta)$. In SSHMC, we use $G_\mathbf{x} = \Sigma^{-1}$ and $G_{(\sigma,\beta)} = \mathbf{I}$. In each ABLA step, the update of $\mathbf{x}$ takes 2 leapfrog steps and the update of $(\alpha, \beta)$ takes 1 leapfrog step. Each SSHMC transition takes 10 ABLA steps. We do not consider HMC on LGCPP, because it mixes extremely slowly for the hyperparameters.

The results of ESS are given in Table 4. The mean of the sampled latent variables and the histogram of sampled hyperparameters are given in Figure 3. It is clear that the samples of RMHMC and SSHMC are consistent, so both methods are mixing well. However, SSHMC generates about *six times* as many effective samples per hour as RMHMC within Gibbs.

## 6 Conclusion

We have presented Semi-Separable Hamiltonian Monte Carlo (SSHMC), a new version of Riemannian manifold Hamiltonian Monte Carlo (RMHMC) that aims to retain the flexibility of RMHMC for difficult Bayesian sampling problems, while achieving greater simplicity and lower computational complexity. We tested SSHMC on several different hierarchical models, and on all the models we considered, SSHMC outperforms both HMC and RMHMC within Gibbs in terms of number of effective samples produced in a fixed amount of computation time. Future work could consider other choices of mass matrix within the semi-separable framework, or the use of SSHMC within discrete models, following previous work in discrete HMC [12, 15].

## Footnotes

[1]Our use of a Gibbs scheme for RMHMC follows standard practice [7].

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
