[Reviews · NeurIPS 2014]

Submitted by Assigned_Reviewer_8

The authors introduce a novel RMHMC type method based on a semi-separable Hamiltonian for use in hierarchical models. The aim of the paper is to enable computationally efficient sampling from hierarchical models where there is strong correlation structure between the parameters and the hyperparameters.

The authors give a clear introduction to hierarchical models and RMHMC before describing the proposed semi-separable integrator.

The “trick” used in this paper is to define metric tensors independently over the parameters and hyperparameters, which allows both sets of parameters to be updated iteratively based on a single Hamiltonian that combines both quantities. Since the metrics used are constant (conditional on the other set of parameters/hyperparameters) it allows the use of the standard leapfrog integrator rather than the computationally more expensive generalised leapfrog integrator.

This is a novel approach for employing constant curvature Riemannian geometry in hierarchical models within an HMC scheme. Clearly for statistical models whose metrics are naturally constant (i.e. derivative of metric is zero), this approach will work very well. The only difficulty is when the parameter metric is non-constant (i.e. varies across the space), since an approximate constant metric must then be estimated for use. The authors make this point already, however I think it is worth making this even clearer in the paper. They suggest for example the use of the metric at the mode of the parameters (for a given set of hyperparameters), however this may be expensive to obtain since it requires an optimisation at each iteration. It would be interesting to see what the impact is on the computational speed if this is necessary at each iteration.

Similarly, for cases with very strong data, the prior curvature at the mode might not be appropriate for describing the posterior geometry of the hyperparameters, if e.g. there is very strong correlation structure in the hyperparameters. It would be interesting to see what sampling would be like in such a case.

A minor gripe about the abstract - this isn’t really a new integrator but rather a clever way of defining the metrics and overall Hamiltonian such that the standard leapfrog integrator can be employed for both parameters and hyperparameters in a combined manner. I would therefore suggest a change from “This structure is exploited by a new integrator which we call the alternating blockwise leapfrog algorithm.” to “This structure is exploited by a new algorithm which we call the alternating blockwise leapfrog algorithm.”

Minor typos:

Page 2, 102, Girolami et al. to Girolami and Calderhead
Page 7, 341 RHMC to RMHMC
Summary: A novel RMHMC approach that allows efficient Bayesian inference over certain classes of hierarchical models.

Submitted by Assigned_Reviewer_15

This paper proposes a way to speed up Hamiltonian Monte Carlo (HMC) sampling for hierarchical models. It is similar in spirit to RMHMC, in which the mass matrix varies according to local topology, except that here the mass matrices for each parameter type (parameter or hyperparameter) only depend on their counterpart, which allows an explicit leapfrog integrator to be used to simulate dynamics rather than an implicit integrator requiring fixed-point iteration to convergence for each step. The authors point out that their method goes beyond straightforward Gibbs sampling with HMC within each Gibbs step since their method leaves the counterpart parameter's momentum intact.

As far as I am aware, the idea is novel. It is also quite interesting, it's presented very clearly and it has some convincing experiments.

Please fix typos and notational inconsistencies (I counted "ALBA" instead of "ABLA" 8 times!).
Summary: The paper presents a novel HMC method, applying tunable mass matrix ideas to hierarchical models without the fuss of implicit integration. It's simple but it seems to work well.

Submitted by Assigned_Reviewer_24

This contribution suggests an approach to addressing the problem of poor mixing in hierarchic Bayesian models. The development of semi-separable RMHMC (ABLA) is the authors way to address the problem of poor mixing in hierarchic models as well as reducing the computational cost of a full blown RMHMC. The main innovation seems to be the choice of a block-diagonal mass matrix however I am a little unclear of the description of this structure as a FIM based mass matrix will have exactly this block diagonal structure as well. My understanding is that the term G_{\theta}(\phi, x) will be a function of *only* \phi and x and like wise for the other component. If I am correct then this looks to be quite an interesting construction for a mass matrix in addressing the correlation of the parameters and hyper-parameters in a hierarchic model. Eveyrthing will decouple as described and if further the mass matrices are fixed then standard Verlet type integrators will suffice providing a computational saving over implicit integrators - at the expense of non-adaptive mixing. The description in 4.3 is nice and intuitive and describes how the suggested mass matrix will be useful for hierarchic models. The reported results illustrate well the improved sampling of the parameters further up in the hierarchy. One wonders then if the suggested block diagonal mass matrix is plugged into RMHMC with adaptive mass for low and high level variables how it would perform?

It is clear that the paper is looking to an RMHMC style solution to the problem of sampling from Bayesian hierarchic models. However there is a vast literature on re-parameyerisation approaches for such models - centred/noncentered, sufficient/ancialiary transformations, recent work by Adams and Murray, and even more recent work using pseudo-marginal schemes by Filiponne and Girolami. It feels that the other half of the evaluation in the paper is missing and the burning question I now have is whether ALBA will be superior or more realistically in what class of models would it be superior to transformations or explicit marginalisation. It is too much to ask the authors to include such comparisons so it might be good of some discussion would be devoted to this issue as it is a question the authors will be plagued with if not addressed at least to some extent.
Summary: A nice contribution which suggests a clever construction of a block-diagonal mass matrix which introduces 'semi-separability' in hierarchic models. For clever choice of this matrix then improved mixing at the higher levels of the model can be realised. This form of mass matrix presumably could be exploited directly in RMHMC achieving exceptional sampling at both low and high levels of the model? Aswith most of these methods the devil is in the detail of devising an appropriate mass matrix of this form and this leads to an interesting issue for further work.

The experiments illustrate the superior mixing at the higher level and are convincing.

It is a pity that the transformation approach or pseudo-marginal approaches to this problem were not covered.

Overall nice job.

Submitted by Assigned_Reviewer_45

The authors introduce a new Hamiltonian and claim that it is better than existing Hamiltonians for heirarchical models. The technical material is very well-written, and the Hamiltonian they present is (as far as I know) novel. It is also very plausible that the new Hamiltonian leads to substantial improvements. However, I found it to be most confusing (and least convincing) in the few places that it argues the new method is better than the old one.
Major Issues:

1. The authors say that they are measuring "effective sample size" but don't say how they are computing this; they reference the reader to [6]. I haven't read [6] carefully, but certainly it doesn't define effective sample size (indeed the article doesn't contain the word "effective" and uses "size" only to refer to actual counts). Normally I wouldn't be so picky, but I think this is critical to their claims. In particular, for 2 of the 3 tables, their proposed method is MUCH slower than the competitors as measured by clock time; they rely on normalized effective sample size to argue that their method is better despite this. I actually think that, almost regardless of the normalization method they use for computing ESS, this is going to be a pretty unconvincing argument (the same table could be used to show that almost any adaptive algorithm is better than almost any other, since adaptive algorithms care so much about initial burn-in periods). But it is especially unconvincing if the normalization is not defined.

This leads me into a bigger reason that their numerics aren't as convincing as I might hope. It is clear from the last column of table 1 (and figure 1) that some of the competitor algorithms are still quite far from stationarity after the number of steps that they run. If each step had a similar cost, that would be a fair comparison: they would be showing that their algorithm converges more quickly, and that's that. The whole variance thing would be a bit of a red herring in this situation (most of the error for the competitors is coming from bias, not variance), but it would be a useful thing to discuss. However, from their own tables, this doesn't seem to be the case. Their algorithm seems to take much more time per step than the competitor algorithms, and they have chosen a NUMBER of STEPS such that their algorithm has converged but the competitors haven't. This really doesn't seem to be a fair comparison, since effective sample size, bias, etc. can all get better as the algorithm runs.

To describe the unfairness a bit more graphically, I'll describe an artifical example that might get something like their table 1. The original algorithm the paper discusses is MCMC driven by kernel $K$ with mixing time $T$. The paper proposes a new algorithm: run $K^{7}$ (that is, take every 7'th point from a Markov chain driven by $K$). We then look at a sample obtained from running both algorithms for $\frac{T}{3}$ steps. Just like Table 3, the new algorithm would take about 7 times as long to run as measured by wall clock. Also just like table 3, we would expect the old algorithm to have enormous bias (it hasn't mixed yet) while the new algorithm would have very tiny bias (it has mixed well). Its clear in my artificial example that the comparison is unreasonable, and I think that for the same reason it is a little unreasonable in this paper.

Minor Issues:

1. Bottom of pp.2, the authors claim that some dynamics are not `valid' because they aren't reversible. They probably meant to say something about how they don't converge to the target.

2. There seems to be a bit of a bait-and-switch at the bottom of page 3. Up until now, the new method has been motivated by pointing out that competitor algorithms scale poorly in high dimensions. When the scaling of this paper's algorithm is revealed, it looks pretty similar for the example under consideration, but this is brushed away as being unimportant. This isn't such a big deal for the applicability of the paper; it just makes the note feel a little dishonest. Alternatively, maybe this is really a big difference, in which case they should explain why.

3. I wish section 4.3 were longer! This seems like the most important section, but it wasn't enough for me to figure out what was going on. This is the sort of section that would be *vastly* improved by *any* toy example where this heuristic could be made precise. In particular, it would be nice to see that there is some optimal value of total energy in the Gibbs version of HMC, and that their method of passing energy back and forth between components (rather than keeping it constant) actually does better than that optimal value. Even just stating the algorithm and values would be enough, which could be done in a few lines. If the claim really is obvious, I hope that they have such an example.

4. In figure 1, I don't think the author's explain the coloring on pictures 2 and 4.
Summary: The authors introduce a new Hamiltonian and claim that it is better than existing Hamiltonians for heirarchical models. The technical material is very well-written, and the Hamiltonian they present is (as far as I know) novel. It is also very plausible that the new Hamiltonian leads to substantial improvements. However, I found it to be most confusing (and least convincing) in the few places that it argues the new method is better than the old one. Also, for hierarchical models, now there are a lot of tools. Thus my question is whether these set of improvements will really make a difference. However, of course, new ideas are always welcome.
Author Feedback
Author rebuttal: Reviewer 15:
Thanks for pointing out the inconsistent notations, we will correct them.

Reviewer 24:

* "the term G_{\theta}(\phi, x) will be a function of *only* \phi and x"
Yes, your understanding is correct.

* “One wonders then if the suggested block diagonal mass matrix is plugged into RMHMC with adaptive mass for low and high level variables how it would perform?”

In the experiments, we consider the same semi-separable mass matrix for SSHMC and RMHMC-within-Gibbs on Gaussian funnel and hierarchical Bayesian logistic regression.

It would be good to compare to RMHMC *without* Gibbs (i.e., RMHMC in the joint space of parameters and hyperparameters), but we didn’t consider in the experiments due to the expensive implicit integrator used by RMHMC. It is likely to converge in fewer iterations but to be too computationally expensive at large scale.

* Recent reparameterization approaches: This is an interesting point.

Manifold HMC methods can be viewed as a local reparameterization technique, but global reparameterizations such as the works your cite would be interesting to consider.

One can imagine combining these reparameterization methods with SSHMC to achieve even better performance.

Reviewer 45:

* Major Issue 1:
The reviewer's concern seems to be that because our method takes more computational cost per sampling iteration, perhaps the competitor methods have not converged in 5000 iterations but would have converged if given the same amount of wall clock time.

This is a reasonable concern, but we have evidence that this is not what is happening in our results. Specifically:

Table 1: Betancourt has provided numerical evidence that standard HMC for this problem is still biased even after 100K iterations. To verify this, we reran our Gaussian funnel experiments, allowing standard HMC and RMHMC within Gibbs to run for 10 times more
iterations. We find that although HMC and RMHMC produce better answers, they still show a large bias, i.e., they do not converge even after 50K iterations, while SSHMC seemed
to converge after only 5k (see Table below).

TABLE (corresponds to Table 1 from paper)

SSHMC result:
ESS of x: 3868.7974
ESS of v: 1541.6733
mean of v: -0.04216*
std of v: 2.9774
Time Taken: 22.1294 s
ESSPS of x: 174.8262
ESSPS of v: 69.6663

HMC result:
ESS of x: 115.3544
ESS of v: 38.9636
mean of v: -0.63157*
std of v: 2.828
Time Taken: 36.6357 s
ESSPS of x: 3.1487
ESSPS of v: 1.0635

RMHMC result:
ESS of x: 1054.3319
ESS of v: 31.6964
mean of v: -1.5821*
std of v: 2.2384
Time Taken: 18.924 s
ESSPS of x: 55.7139
ESSPS of v: 1.6749

where, ESSPS is ESS per second, x is the low-level parameters and v is the hyper-parameter.
The true value of hyper-parameter is 0.
* marks the mean of sampled hyperparameter.

Table 3: Figures 2 and 3 show samples of the hyperparameter values in this model, which appear to all have approximately converged to the same distribution. This suggests that an ESS comparison is appropriate. It is worth mentioning that our results from the competitors were given by reproducing the experiments in the original paper of RMHMC [9] with all settings of the number of burin-in iterations, RMHMC parameters, etc.

* Minor Issue 1:
Deterministic updates (like Hamiltonian dynamics) must be reversible to be valid.

* Minor Issue 2:
This is not a bait-and-switch. Our algorithm scales much better than RMHMC *without* Gibbs (i.e., RMHMC jointly in the parameter and hyperparameter space). Indeed, RMHMC without Gibbs is infeasible for all our examples except the first toy example. Our algorithm does scale similarly to RMHMC *within* Gibbs, but as we show in the experiments, SSHMC mixes much better. The goal of our paper is to get the effectiveness of joint-RMHMC but the scalability of RMHMC-within-Gibbs.

* Minor Issue 3:
We will consider the suggestion of extending Section 4.3.

* Minor Issue 4:
We will add legend for the second and fourth plots in Figure 1.

* Reviewer 8:
Although we suggested using the metric at the mode as a possible choice, this is not the choice that we used in the experiments, precisely to avoid the iterative approach, as the reviewer points out.

For example, in the Bayesian hierarchical logistic regression experiment,
we used a simple approximation of Hessian given by
H_{approx}(\theta) = \sum_i x_i x_i^T + vI,
where \theta is the parameter in logistic function, x_i is the input feature of data point i, and v is the variance hyperparameter in the prior.

It is remarkable that, although this is an admittedly crude approximation, this was still good enough to obtain better performance than the competitors.

* “There is very strong correlation structure in the hyperparameters. It would be interesting to see what sampling would be like in such a case”

In our experiments, the hyperparameters of the Log-Gaussian Cox model are indeed strongly correlated, but our method appears to perform much better than RMHMC-within-Gibbs. So, we would believe that the correlation in hyperparameters would not affect the mixing speed of our method much.

We will consider the change in the abstract and correct typos.